REGISTERED REPORT PROTOCOL

# Effects of using wearable devices to monitoring physical activity in pulmonary rehabilitation programs for chronic respiratory diseases: A systematic review protocol

**Thaianne Rangel Agra Oliveira**[1]ᵒ*, **Ana Tereza do Nascimento Sales Figueiredo Fernandes**[2]ᵒ, **Thayla Amorim Santino**[2]ᵒ, **Fernanda Elizabeth Pereira da Silva Menescal**[3]ᵒ, **Patrícia Angélica de Miranda Silva Nogueira**[1]ᵒ

**1** Departament of Physical Therapy, Federal University of Rio Grande do Norte, Natal, RN, Brazil, **2** Departament of Physical Therapy, State University of Paraíba, Campina Grande, PB, Brazil, **3** School of Physical & Occupational Therapy, McGill University Montreal, Montreal, Québec, Canada

ᵒ These authors contributed equally to this work.
* thaianne.agra.063@ufrn.edu.br

## Abstract

### Introduction

Pulmonary rehabilitation (PR) is an intervention aimed at the comprehensive care of individuals with chronic respiratory diseases. Patients with chronic obstructive pulmonary disease (COPD) and asthma present low levels of physical fitness because they avoid physical exercises due to the fear of triggering recurrent symptoms. Wearable devices have been integrated into behavioral modification interventions for physical activity in PR protocols. Therefore, this review aims to identify how wearable devices are being utilized for monitoring chronic respiratory diseases in pulmonary rehabilitation programs.

### Methods and analysis

Searches will be conducted on Medline, Cochrane Central Register of Controlled Trials, Embase (CENTRAL), CINAHL and PEDro electronic databases, as well as a search in the grey literature. We will include baseline data from randomized clinical trials reporting the use of wearable devices for monitoring physical activity in protocols for pulmonary rehabilitation programs for chronic respiratory diseases. Studies that discuss only the development of algorithms or applications for the assessment of diseases or unavailable full texts will be excluded. The main reviewer will conduct the initial search and exclusion of duplicates, while two independent reviewers will select studies, extract data, and assess the methodological quality using the PEDro tool.

### PROSPERO registration number

CRD42024504137.

**Data Availability Statement:** All relevant data are within the manuscript and its Supporting Information files.

**Funding:** The author(s) received no specific funding for this work.

**Competing interests:** The authors have declared that no competing interests exist.

## Introduction

Pulmonary rehabilitation (PR) is an intervention focused on the comprehensive care of individuals with chronic respiratory diseases (CRD), encompassing a thorough assessment coupled with therapies tailored to the individuality of each patient, including physical and aerobic training, health education, and behavior change [1, 2].

CRD refers to diseases of the airways and other structures of the lungs, accounting for approximately 7.5 million deaths per year, roughly 14% of annual deaths worldwide [3, 4]. Among these diseases, chronic obstructive pulmonary disease (COPD) stands out as the third leading cause of global mortality [5] and asthma, recognized as a serious public health issue, is associated with increased morbidity and mortality [6, 7].

Patients with COPD tend to be less physically active, resulting in loss of muscle mass and function, coupled with a reduction in exercise capacity, limiting factors that can impact social interaction and independence [2]. Individuals with asthma exhibit low levels of physical fitness as they tend to avoid physical exercises due to the fear of triggering recurrent symptoms such as wheezing, dyspnea and cough. This avoidance leads to dysfunction of peripheral muscles, reducing functional capacity and quality of life [8].

Thus, it can be asserted that physical inactivity is associated with negative outcomes in patients with CRD, including alterations in pulmonary function, hospitalization and mortality [9]. Physical activity (PA) consists of any bodily movement produced by the contraction of skeletal muscles that increases energy expenditure above the basal level [10]. Traditionally, measures of physical performance are assessed subjectively or objectively, including evaluations of mobility, balance and strength. However, these assessments may be limited by the influence of behavioral changes and the lack of representativeness of the individual's actual performance in their daily context [11]. Therefore, there is consensus that PA monitors are more accurate tools for directly measuring the PA outcome, rather than questionnaires [12].

With the advancement of technology, it is possible to monitor and record real-time information about physical performance through the capture of data such as PA levels, mobility and physiological patterns from wearable devices [13]. Health monitoring systems through these devices involve various types of flexible sensors that can be integrated into textile fibers, clothing, elastic bands or directly attached to the human body [14]. Thus, concerning the scope of PR, new behavior change techniques to promote activity through technological developments have shown promise in this scenario [15].

Although PR is crucial for improving exercise capacity and quality of life in individuals with CRD, there are still challenges in inserting and progressing daily PA into the routine of these patients [16]. In this sense, validated activity monitors have been necessary to evaluate the effects of PR on monitoring the PA of these patients [1]. For this reason, behavioral modification interventions for PA have been implemented and associated with daily monitoring and feedback through the use of wearable devices integrated into PR protocols [16].

Given the above, the aim of the present study is to analyze the literature to identify how wearable devices are being utilized for monitoring chronic respiratory diseases in pulmonary rehabilitation programs. Likewise, it aims to define which parameters obtained by the devices have been most commonly used and how they have impacted behavioral modification of PA, consequently contributing to the improvement of the quality of life of these patients.

## Methods and analysis

### Registry

This protocol was elaborated according to the Preferred Reporting Items for Systematic Reviews and Meta-Analyses Protocols (PRISMA-P) [17] (S1 Checklist) and registered in the International Prospective Registry of Systematic Reviews (PROSPERO) with registration number CRD42024504137 in January 2024.

### Eligibility criteria

**Types of study.**   Randomized clinical trials will be included, without restrictions on language and year of publication. Studies with full text unavailable and studies that discuss only the development of algorithms or applications for the monitoring of chronic respiratory diseases will be excluded. In the case of articles written in languages the authors do not master, professional translation services will be hired to interpret and extract data from these articles.

**Types of participants.**   Participants must be subjects of both sexes aged 18 and over, with a clinical diagnosis of COPD or asthma.

**Exposition.**   Studies utilizing wearable devices for monitoring in pulmonary rehabilitation programs for chronic respiratory diseases, including resistance and/or aerobic physical training will be considered. Wearable devices monitor and record real-time information about an individual's physiological condition and movement activities. They are composed of electronic sensors that can be integrated into textile fibers, clothing, elastic bands, or directly attached to the human body. The most commonly used sensors are wireless inertial sensors, identified in the form of accelerometers, gyroscopes and pedometers, which provide information on the intensity and duration of physical activity [9]. Wearable devices that provide quantitative measures related to mobility, physical activity and/or physiological measures, capable of monitoring physical performance will be considered for the development of this study.

### Types of outcome measures

**Primary outcomes.**

1. Health-related quality of life (eg. measured by specific or generic questionnaires like Short Form Survey SF-36, Clinical COPD Questionnaire and The Asthma Quality of Life Questionnaire (AQLQ).

2. Physical activity (eg. number of steps recorded by wearable devices or measured by self-reported physical activity questionnaires. For analysis purposes, this outcome will be standardized using the standard mean difference (SMD) in the meta-analysis. If a meta-analysis is not feasible, the difference between pre and post-intervention measurements will be calculated and the percentage of this difference will be computed, standardizing the data to a common unit of measurement).

3. Functional capacity (eg. measured by submaximal exercise tests, such as Six-minute Walk Test (6MWT), Six-minute Step Test (6MST), Incremental Shuttle Walk Test (ISWT).

4. Adherence to rehabilitation.

**Secondary outcomes.**

1. Peripheral muscle strength (one-repetition maximum (1RM).

2. Respiratory muscle strength (manovacuometry).

3. Dyspnea/ symptoms (eg. measured by questionnaires, such as Modified Medical Research Council (mMRC), COPD assessment test (CAT), asthma control questionnaire (ACQ).

4. Anxiety (eg. measured by specific or generic questionnaires, like Hospital Anxiety and Depression Scale (HADS).

## Search

**Search strategy.**   Searches will be conducted in Medline, Cochrane Central Register of Controlled Trials (CENTRAL), Embase, CINAHL and PEDro databases, as well as a search in the gray literature. Full-text studies will be selected with no language or publication year restrictions. The search strategy will be constructed using terms related to the population (COPD and asthma) and intervention (exercise therapy, pulmonary rehabilitation, exercise program, wearable devices, smartwatch), the search terms will be combined using the boolean operators OR or AND. The search strategy for Medline can be found in the S1 File. Additionally, a manual search will be conducted in the reference lists of included studies. The articles will be included after reviewing the titles, abstracts and full texts.

**Study selection.**   After identifying the studies, all files will be transferred to the Mendeley reference manager (https://www.mendeley.com) to import the results and remove the duplicates. We will export the reference list to the Ryyan QCRI systematic reviews web-based application (https://rayyan.qcri.org) for title and abstract selection by the main reviewer (TRAO). Two additional reviewers (ATdNSFF and TAS), blinded to each other, will independently read the titles and abstracts, and selection by these reviewers will be matched with the selection of the lead reviewer. In case of disagreement a third reviewer (PAdMSN) will be consulted to reach a consensus considering the previously established eligibility criteria. The eligible articles will be obtained in full text. The detailed selection process of the search will be recorded for the development of a flowchart in accordance with the PRISMA guidelines.

**Data extraction.**   The following data will be extracted by a reviewer using a standardized form and verified by a second reviewer:

1. Participants (age, sex, health conditions);

2. Methods (study design, sample size, country and year);

3. Sensor used and location in the human body;

4. Protocol used in the pulmonary rehabilitation program (intervention, comparison, duration of intervention, frequency of intervention);

5. Results (primary and secondary outcomes);

6. Limitations of the study.

Extraction data from included studies will be done independently by two reviewers (TRAO and (FEPdSM). The outcomes not reported will be signalized in the 'Characteristics of included studies'.

**Quality assessment.**   The PEDro scale assessment tool will be applied by two independent reviewers (TRAO and PAdMSN) to evaluate the methodological quality of the selected studies. The following scale criteria will be used: eligibility criteria, random allocation, concealed allocation, baseline comparability, blinding of participants, blinding of therapists, blinding of assessors, adequate follow-up, intention-to-treat analysis, between group comparisons and measures of variability. The total scores of the scale range from 0 to 10, and risk of bias will be interpreted as high (0–3), moderate (5–7) or low (8–10) [18]. Results will

be presented in a table containing the name of authors and scores obtained on each item of the PEDro Scale.

The assessment of uncertainty around the evidence will be conducted through the Grading of Recommendations Assessment, Development and Evaluation (GRADE) system. Three authors (TRAO, ATdNSFF and TAS) will use the five grading of GRADE considerations (risk of bias, consistency of effect, imprecision, indirectness and publication bias) to assess the quality of evidence as it relates to the studies that contribute data for the prespecified outcomes.

**Analysis.**   The results of each study will be summarized in tables for the elaboration of a quantitative synthesis of the study. The meta-analysis will be conducted if the necessary data are made available regarding the primary and/or secondary outcomes. If necessary, subgroup analyses may be conducted to observe differences in outcomes among chronic respiratory diseases. To evaluate the effect size (Z) a value of $P<0.05$ will be considered and for heterogeneity ($I^2$) a value of $P<0.10$ [19].

**Study timeline.**   This review will be conducted for 8 to 12 months.

## Patient or public involvement

They will be involved in the dissemination stage of the results for the general population, in the form of lay summaries that can be posted on social media.

## Ethics and disclosure

No previous ethical approval is required for this review. The findings of this review will be submitted to a scientific journal, disclosed at international scientific events, and shared in social media using accessible language.

## Discussion

The utilization of wearable devices in monitoring physical activity within pulmonary rehabilitation programs for chronic respiratory diseases represents an innovative and promising approach in the management of these health conditions. This practice integrates technology into the fields of medicine and rehabilitation, aiming to enhance the effectiveness of treatment programs and optimize outcomes for patients with CRD, such as COPD and asthma.

In this regard, the present study aims to provide a systematic review offering a comprehensive analysis of the effectiveness and impact of these devices in this specific context. It primarily contributes to the assessment of clinical efficacy, technological integration into rehabilitation programs, improvement in exercise adherence, evidence synthesis, and identification of gaps and future research needs.

Therefore, the use of wearable devices in monitoring physical activity within PR for CRD represents a novel method that has the potential to transform the management of these conditions, providing a more comprehensive, personalized, and effective treatment perspective. However, it is crucial to address ethical challenges, scientifically validate these technologies, and promote accessibility to maximize their benefits.

## Supporting information

**S1 Checklist. PRISMA-P 2015 checklist.**
(DOCX)

**S1 File. Search strategy for Medline.**
(DOCX)

## Author Contributions

**Conceptualization:** Thaianne Rangel Agra Oliveira, Ana Tereza do Nascimento Sales Figueiredo Fernandes, Thayla Amorim Santino, Fernanda Elizabeth Pereira da Silva Menescal, Patrícia Angélica de Miranda Silva Nogueira.

**Methodology:** Thaianne Rangel Agra Oliveira, Ana Tereza do Nascimento Sales Figueiredo Fernandes, Thayla Amorim Santino.

**Writing – original draft:** Thaianne Rangel Agra Oliveira, Ana Tereza do Nascimento Sales Figueiredo Fernandes, Thayla Amorim Santino.

**Writing – review & editing:** Thaianne Rangel Agra Oliveira, Ana Tereza do Nascimento Sales Figueiredo Fernandes, Thayla Amorim Santino, Fernanda Elizabeth Pereira da Silva Menescal, Patrícia Angélica de Miranda Silva Nogueira.

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
