## [Decision Letter · Decision Letter 0]

20 Jun 2024

PONE-D-24-10390Effects of using wearable devices to monitoring physical activity in pulmonary rehabilitation programs for chronic respiratory diseases: a systematic review protocolPLOS ONE

Dear Dr. Oliveira,

Thank you for submitting your manuscript to PLOS ONE. After careful consideration, we feel that it has merit but does not fully meet PLOS ONE’s publication criteria as it currently stands. Therefore, we invite you to submit a revised version of the manuscript that addresses the points raised during the review process.

**It is worth noting that authors should provide more detailed information about the included studies, as adequate and high-quality research is a prerequisite for the meta-analyses.**

We look forward to receiving your revised manuscript.

Kind regards,

Yongzhong Guo, Ph.D

Academic Editor

PLOS ONE

Journal Requirements:

2. In your cover letter, please confirm that the research you have described in your manuscript, including participant recruitment, data collection, modification, or processing, has not started and will not start until after your paper has been accepted to the journal (assuming data need to be collected or participants recruited specifically for your study). In order to proceed with your submission, you must provide confirmation.

Reviewers' comments:

Reviewer's Responses to Questions

**Comments to the Author**

1. Does the manuscript provide a valid rationale for the proposed study, with clearly identified and justified research questions?

Reviewer #1: Yes

Reviewer #2: Yes

2. Is the protocol technically sound and planned in a manner that will lead to a meaningful outcome and allow testing the stated hypotheses?

Reviewer #1: Yes

Reviewer #2: Partly

3. Is the methodology feasible and described in sufficient detail to allow the work to be replicable?

Reviewer #1: Yes

Reviewer #2: Yes

4. Have the authors described where all data underlying the findings will be made available when the study is complete?

Reviewer #1: Yes

Reviewer #2: Yes

5. Is the manuscript presented in an intelligible fashion and written in standard English?

Reviewer #1: Yes

Reviewer #2: Yes

6. Review Comments to the Author

You may also provide optional suggestions and comments to authors that they might find helpful in planning their study.

**Reviewer #1: **A practical and justifiable review of literature which should shed light on role of wearables in pulmonary rehabilitation programmes.

**Reviewer #2:** Rangel Agra Oliveira et al present a planned systematic review of the effects of using wearable devices in monitoring physical activity in pulmonary rehabilitation programs for chronic respiratory diseases. Given the nearly ubiquitous nature of wearable health devices, this is a timely and necessary systematic review. The proposed review protocol appears comprehensive and well-developed. There are two minor concerns:

1. There is currently no limitation on language of studies, but the authors do not comment on how they plan on interpreting and extracting papers that are written in languages for which they do not personally have mastery. This should be clearly delineated, or a language restriction should be considered.

2. The authors should better delineate how they plan to standardize and quantify primary outcome #2 physical activity, particularly for self-reported physical activity level.

7. PLOS authors have the option to publish the peer review history of their article (what does this mean?). If published, this will include your full peer review and any attached files.

Reviewer #1: **Yes: **Dr Manu Chopra

Reviewer #2: No

---

## [Author Response · Author response to Decision Letter 0]

2 Jul 2024

Dear Editors and reviewers,

We would like to express our sincere gratitude to you and the reviewers for the careful and insightful review of our manuscript titled “Effects of Using Wearable Devices to Monitoring Physical Activity in Pulmonary Rehabilitation Programs for Chronic Respiratory Diseases: A Systematic Review Protocol.” We appreciate the constructive feedback provided, which has significantly contributed to improving the quality and clarity of our work. Below, we address each of the points raised by the reviewers.

Journal Requirements:

1. The manuscript was formatted according to PLOS ONE style requirements, including those for file naming.

2. Was added to cover letter that: “ the progress of study has not started and will not begin until the article is accepted by the journal”.

3. We confirm that “All relevant data are within the manuscript and its Supporting Information files”. The data that will be used for the analyzes will be made available in supplementary material when submitting the complete work.

Reviewers’ comments:

1. In the case of articles written in languages the authors do not master, professional translation services will be hired to interpret and extract data from these articles.

2. Physical activity (eg. number of steps recorded by wearable devices or measured by self-reported physical activity questionnaires. For analysis purposes, this outcome will be standardized using the standard mean difference (SMD) in the meta-analysis. If a meta-analysis is not feasible, the difference between pre and post-intervention measurements will be calculated and the percentage of this difference will be computed, standardizing the data to a common unit of measurement).

Thank you once again for your time and consideration.

Sincerely,

Thaianne Rangel Agra Oliveira

---

## [Decision Letter · Decision Letter 1]

17 Jul 2024

Effects of using wearable devices to monitoring physical activity in pulmonary rehabilitation programs for chronic respiratory diseases: a systematic review protocol

PONE-D-24-10390R1

Dear Dr. a Oliveira,

We’re pleased to inform you that your manuscript has been judged scientifically suitable for publication and will be formally accepted for publication once it meets all outstanding technical requirements.

Kind regards,

Yongzhong Guo, Ph.D

Academic Editor

PLOS ONE

Additional Editor Comments (optional):

Reviewers' comments:

Reviewer's Responses to Questions

**Comments to the Author**

1. Does the manuscript provide a valid rationale for the proposed study, with clearly identified and justified research questions?

Reviewer #1: Yes

Reviewer #2: Yes

2. Is the protocol technically sound and planned in a manner that will lead to a meaningful outcome and allow testing the stated hypotheses?

Reviewer #1: Yes

Reviewer #2: Yes

3. Is the methodology feasible and described in sufficient detail to allow the work to be replicable?

Reviewer #1: Yes

Reviewer #2: Yes

4. Have the authors described where all data underlying the findings will be made available when the study is complete?

Reviewer #1: Yes

Reviewer #2: Yes

5. Is the manuscript presented in an intelligible fashion and written in standard English?

Reviewer #1: Yes

Reviewer #2: Yes

6. Review Comments to the Author

You may also provide optional suggestions and comments to authors that they might find helpful in planning their study.

Reviewer #1: The research question addresses a common practical intervention in pulmonary rehabilitation and aims to contribute to it's real usefulness scientifically. The manuscript thoroughly describes the methods, ensuring transparency and reproducibility by detailing the procedures and analysis pipeline. This includes necessary controls and a statistical power analysis, where relevant. The methodology appears feasible and sufficiently detailed for replication. The authors have committed to making all underlying data available as per the PLOS Data policy. The manuscript is clearly written in standard English, with minimal typographical or grammatical errors.

Reviewer #2: Rangel Agra Oliveira et al re-submit a planned systematic review of the effects of using wearable devices in monitoring physical activity in pulmonary rehabilitation programs for chronic respiratory diseases. Prior comments regarding handling of literature written in languages for which they do not have mastery and plans for standardizing physical activity levels have been addressed. Minor grammatical errors remain, e.g. "in the cases of articles written in languages the authors do not master," "participants must be subjects of both sexes," "the findings of this review will be submitted to a peer-reviewed journal, discloses at international scientific conferences," and ought to be corrected prior to publication for clarity.

7. PLOS authors have the option to publish the peer review history of their article (what does this mean?). If published, this will include your full peer review and any attached files.

Reviewer #1: **Yes: **Dr Manu Chopra

Reviewer #2: No

---

## [Editor Report · Acceptance letter]

19 Jul 2024

PONE-D-24-10390R1 

PLOS ONE

Dear Dr. Oliveira, 

I'm pleased to inform you that your manuscript has been deemed suitable for publication in PLOS ONE. Congratulations! Your manuscript is now being handed over to our production team.

Kind regards, 

on behalf of

Dr. Yongzhong Guo 

Academic Editor

PLOS ONE